# Increased Hepatic *ATG7* mRNA and ATG7 Protein Expression in Nonalcoholic Steatohepatitis Associated with Obesity

**DOI:** 10.3390/ijms24021324

**Published:** 2023-01-10

**Authors:** Andrea Barrientos-Riosalido, Monica Real, Laia Bertran, Carmen Aguilar, Salomé Martínez, David Parada, Margarita Vives, Fàtima Sabench, David Riesco, Daniel Del Castillo, Cristóbal Richart, Teresa Auguet

**Affiliations:** 1Grup de Recerca GEMMAIR (AGAUR)-Medicina Aplicada (URV), Departament de Medicina i Cirurgia, Universitat Rovira i Virgili (URV), Institut d’Investigació Sanitària Pere Virgili (IISPV), 43007 Tarragona, Spain; 2Servei Medicina Interna, Hospital Universitari Joan XXIII Tarragona, Mallafré Guasch, 4, 43007 Tarragona, Spain; 3Servei Anatomia Patològica, Hospital Universitari Joan XXIII Tarragona, Mallafré Guasch, 4, 43007 Tarragona, Spain; 4Servei Anatomia Patològica, Hospital Universitari Sant Joan de Reus, Avinguda Doctor Josep Laporte, 2, 43204 Reus, Spain; 5Servei de Cirurgia, Hospital Sant Joan de Reus, Departament de Medicina i Cirurgia, URV, IISPV, Avinguda Doctor Josep Laporte, 2, 43204 Reus, Spain

**Keywords:** ATG7, NAFLD, autophagy, nonalcoholic steatohepatitis, lipid metabolism

## Abstract

The autophagy gene *ATG7* has been shown to be essential for the induction of autophagy, a process that used to be suppressed in nonalcoholic fatty liver disease (NAFLD). However, the specific role of ATG7 in NAFLD remains unclear. The aim of this study was to analyze hepatic *ATG7* mRNA and ATG7 protein expression regarding obesity-associated NAFLD. Patients included women classified into normal weight (NW, *n* = 6) and morbid obesity (MO, *n* = 72). The second group was subclassified into normal liver (NL, *n* = 11), simple steatosis (SS, *n*= 29), and nonalcoholic steatohepatitis (NASH, *n* = 32). mRNA expression was analyzed by RT–qPCR and protein expression was evaluated by Western blotting. Our results showed that NASH patients presented higher *ATG7* mRNA and ATG7 protein levels. *ATG7* mRNA expression was increased in NASH compared with SS, while ATG7 protein abundance was enhanced in NASH compared with NL. *ATG7* mRNA correlated negatively with the expression of some hepatic lipid metabolism-related genes and positively with endocannabinoid receptors, adiponectin hepatic expression, and omentin levels. These results suggest that ATG7-mediated autophagy may play an important role in the pathogenesis of NAFLD, especially in NASH, perhaps playing a possible protective role. However, this is a preliminary study that needs to be further studied.

## 1. Introduction

The global incidence of nonalcoholic fatty liver disease (NAFLD) is estimated to be 25%, which has been increasing in recent years in parallel with the obesity epidemic. Thus, NAFLD has become the most prevalent chronic liver disease worldwide [1]. NAFLD and metabolic syndrome are highly related since obesity and type 2 diabetes mellitus (T2DM), two metabolic syndrome comorbidities, are the main risk factors for NAFLD [2]. In this sense, NAFLD incidence in patients presenting morbid obesity (MO) can be as high as 90%, whereas it is approximately 70% in diabetic subjects [3,4].

The most important problem of NAFLD, from the hepatic point of view, is simple steatosis (SS), excess fat accumulated in the liver in >5% of hepatocytes; although it can be reversible and can evolve into nonalcoholic steatohepatitis (NASH), fibrosis, and cirrhosis, which are major causes of liver transplantation [5,6]. Given that NASH can be accompanied by fibrosis, the main risk factor for mortality due to hepatic causes of NAFLD [7], in recent years, there has been significant research into therapies aimed at controlling NASH. However, no Food and Drug Administration (FDA)-approved NAFLD-specific medications are currently available [8]. For this reason, it is very important to insist on the study of the pathogenic mechanisms involved in the progression of NAFLD/NASH to demonstrate possible therapeutic targets.

During NAFLD development, multiple cellular interactions occur in hepatic tissue involving hepatocytes, hepatic stellate cells (HSCs), and macrophages or Kupffer cells (KCs) [9,10]. On the one hand, the steatosis process leads to lipotoxicity in hepatocytes, which secrete exosomes to activate HSCs [11]. On the other hand, a high-fat diet (HFD), obesity, and NAFLD itself are related to the presence of intestinal dysbiosis. Dysbiosis involves a change in the gut microbiome that induces an increase in intestinal barrier permeability, which permits the transfer of bacterial endotoxins to the liver through portal vein circulation [12]. Once in the liver, endotoxins activate KCs and promote the release of proinflammatory cytokines, such as interleukin (IL)-6, IL-1, the tumor necrosis factor (TNF)-α, and profibrotic mediators, such as the transforming growth factor (TGF)-β [13], which activate HSCs.

Autophagy is an intracellular degradation system of damaged organelles and misfolded proteins that is important for the maintenance of cell homeostasis. Autophagy inhibits apoptosis and promotes cell survival. The dysregulation of autophagy is associated with the development of metabolic syndrome, hepatic disorders, some pulmonary, renal, and infectious diseases, cardiovascular disease, neurodegenerative disorders, and cancer [14,15]. Recently, autophagy has been reported to be involved in the degradation of lipid droplets in hepatocytes [16]. In this regard, a potential protective role of this process in fatty liver diseases could be suggested since autophagy in hepatocytes has been shown to be suppressed in NAFLD [17]. For instance, Tanaka et al. stated that this process is suppressed in the liver via the induction of the autophagy-inhibiting protein Rubicon [18]. On the other hand, hepatic nonparenchymal cells, including KCs and HSCs, may also use autophagy to maintain their homeostasis or function, thus affecting proinflammatory and profibrotic responses in NAFLD progression [19].

Given the suggested beneficial role that autophagy seems to have in NAFLD pathogenesis and given that there is not yet a specific drug approved by regulatory agencies to treat this hepatic disorder, drugs involving the autophagy system have begun to be investigated for the treatment of NAFLD [8,20,21,22]. In a recent study in animal models, it was suggested that the endocannabinoid system, which is known to be implicated in the pathogenesis of NAFLD, might mediate its role through autophagy mechanisms [23]. In addition, recent findings on the precise mechanism of autophagy regulation in NAFLD have demonstrated that several molecules, such as cluster of differentiation 36, leucine aminopeptidase 3, and severe acute respiratory syndrome coronavirus 2 (SARS-CoV-2) spike protein, act as negative regulators of autophagy. However, thyroid hormone, hydrogen sulfide, melatonin, DA-1241 (a novel G protein-coupled receptor 119 agonist), vacuole membrane protein 1, nuclear factor erythroid 2-related factor 2, sodium glucose cotransporter type-2 inhibitors, immunity-related GTPase-M, and autophagy-related gene 7 (*ATG7*) induce autophagy [24].

Specifically, the *ATG7* gene encodes an E1-like activating enzyme that is essential for autophagy and the transport of cytoplasmic vacuoles. The encoded protein is thought to modulate p53-dependent cell cycle pathways during prolonged metabolic stress. Its action has been associated with multiple functions, including axon membrane trafficking, axonal homeostasis, mitophagy, adipose differentiation, and hematopoietic stem cell maintenance [25].

Recently, in a study that sequenced the complete exosome of 310 patients with NAFLD, rare mutations in the *ATG7* gene were found. These mutations lead to an increased risk of developing severe liver disease in patients with dysfunctional metabolism. Additionally, these mutations caused an alteration in protein function, inducing impairment of cellular content self-renewal and leading to liver damage and inflammation [26]. On the other hand, a study in skeletal muscle-specific *Atg7* knockout mice exhibited reduced lipid accumulation and increased oxidation-related gene expression. In addition, HFD-fed *Atg7* knockout mice presented decreased expression of hepatic lipogenic genes and were protected against diet-induced obesity and insulin resistance [27]. In this context, although ATG7, a main regulator of autophagy, seems to play a protective role in NAFLD pathogenesis (Figure 1), this mechanism remains unclear and needs to be further studied.

Therefore, in the present study, we wanted to explore the role of ATG7 in NAFLD associated with obesity by studying the *ATG7* mRNA and the protein expression of ATG7 in the liver of normal-weight subjects (the control group) and a cohort with MO and different degrees of liver involvement. In addition, we investigated the association between the hepatic mRNA expression of *ATG7* and the main lipid metabolism-related genes and some pro- and anti-inflammatory molecules in the context of NAFLD.

## 2. Results

### 2.1. Baseline Characteristics of Subjects

According to their body mass index (BMI), women were divided into two groups: those who had a normal weight (NW, BMI < 25 kg/m^2^, *n* = 6) and those who presented MO (BMI ≥ 40 kg/m^2^, *n* = 72). Then, subjects with MO were subclassified in accordance with the hepatic histology as normal liver (NL, *n* = 11), SS (*n* = 29), and NASH (*n* = 32). Table 1 shows the cohort’s clinical characteristics and biochemical measurements. The groups were comparable in terms of glycosylated hemoglobin (HbA1c), cholesterol, high-density lipoprotein cholesterol (HDL-C), low-density lipoprotein cholesterol (LDL-C), aspartate-aminotransferase (AST), alanine-aminotransferase (ALT), gamma-glutamyltransferase (GGT), and alkaline phosphatase (ALP). In the analysis, we found that the group with NW, apart from having a lower weight and BMI, also had lower insulin levels than the MO group; in addition, we reported a lower value for the homeostatic model assessment method–insulin resistance (HOMA1-IR) in NW subjects compared to the SS and NASH groups. Then, increased levels of triglycerides (TGs) in the SS group in comparison with NL patients were reported, and we observed enhanced levels of systolic blood pressure (SBP), diastolic blood pressure (DBP), and glucose in NASH subjects compared to NL subjects. Finally, we found that DBP levels were enhanced in NASH compared to SS subjects.

### 2.2. Evaluation of the Relative mRNA and Protein Expressions of Hepatic ATG7 According to BMI

Since ATG7 has been reported to play a key role in autophagy and this process is known to aid in the removal of lipid droplets in the liver, we evaluated ATG7 expression in liver samples in a cohort of women with or without MO presenting different degrees of hepatic involvement.

First, we analyzed the hepatic relative *ATG7* mRNA and ATG7 protein expression in a cohort that was classified according to whether they presented NW or MO. In this analysis, we did not find significant differences between groups (Figure 2A,B).

### 2.3. Evaluation of the Relative mRNA and Protein Expressions of Hepatic ATG7 According to Liver Histology

Then, we wanted to evaluate the relative hepatic ATG7 mRNA and ATG7 protein expression when subjects were divided based on NAFLD or NASH presence. Regarding the cohort classified by the presence or absence of NAFLD, we did not observe significant differential expression between groups, neither in the mRNA analysis nor in the protein analysis (Figure 3A,B). However, we found higher levels of hepatic ATG7 mRNA and ATG7 protein expression in the NASH cohort than in non-NASH patients (Figure 3C,D, respectively).

Later, we subclassified the patients with MO according to their hepatic histopathological degree into NL, SS, or NASH, to explore the main topic of this study, the role of ATG7 in NAFLD stages. In this sense, we observed higher expression of ATG7 mRNA in the NASH group than in the SS cohort, as shown in Figure 4A. In addition, we reported enhanced expression of ATG7 protein in NASH subjects compared to NW women (Figure 4B).

### 2.4. Evaluation of the Relative mRNA and Protein Expressions of Hepatic ATG7 According to NASH-Related Parameters

Since our previous results demonstrated a higher expression of ATG7/ATG7 in NASH, we wanted to focus the analysis on the main parameters related to this advanced stage of NAFLD, such as portal and lobular inflammation, and hepatocellular ballooning. In this regard, we observed an increase in ATG7 mRNA and ATG7 protein expression in patients presenting inflammation compared to those without inflammation (*p* = 0.017 and *p* = 0.039, respectively).

Then, when we classified inflammation as portal or lobular, we reported an enhanced hepatic ATG7 mRNA and ATG7 protein expression in patients with mild lobular inflammation compared to those with no inflammation (*p* = 0.024 and *p* = 0.015, respectively). Concerning other NASH-related parameters, such as hepatic ballooning or portal inflammation, we did not observe differential expression of ATG7/ATG7 between groups.

### 2.5. Evaluation of the Relative mRNA and Protein Expressions of ATG7 According to NAFLD Comorbidities

Since autophagy mediated by ATG7 is often disrupted in metabolic dysfunction, we wanted to study hepatic *ATG7* mRNA and ATG7 protein expression in patients classified according to the presence of comorbidities associated with NAFLD (dyslipidemia, T2DM, hypertension, and metabolic syndrome) other than obesity. In this sense, we only found higher *ATG7* mRNA expression in the presence of dyslipidemia (*p* = 0.005) but not in the protein analysis. Nevertheless, we did not find significant differences concerning the other NAFLD comorbidities.

### 2.6. Correlations between Relative mRNA and Protein Expressions of Hepatic ATG7, with Clinical and Biochemical-Related Parameters

Finally, to deepen our understanding of ATG7 in NAFLD associated with obesity, we examined its relationships with the levels of several clinical parameters and metabolic and inflammatory mediators. Regarding *ATG7* mRNA expression, we found a negative correlation with the hepatic mRNA expression of carnitine palmitoyl transferase deficiency-type 1 (*CPT1a*), liver X receptor alpha (*LXRα*), and retinol transporter protein type 4 (*RBP4*). However, we observed a positive correlation between the mRNA expression of cannabinoid receptors (*CB1* and *CB2*) and adiponectin. All of these correlations are graphically represented in Figure 5A–F. Unfortunately, we did not report significant associations with other hepatic lipid metabolism-related genes, such as sterol regulatory element-binding protein 1c (*SREBP1c*), fatty acid synthase (*FAS*), fatty acid-binding proteins (*FABP*), or with other pro- and anti-inflammatory ILs.

## 3. Discussion

In the present study, we explored the role of ATG7 in the pathogenesis of NAFLD by evaluating the hepatic *ATG7* mRNA and ATG7 protein expression in a cohort of patients with NAFLD associated with MO using NW subjects and MO individuals, with NL as control groups. In addition, we analyzed whether there is any association between the hepatic expression of ATG7 and the main lipid metabolism-related genes and inflammatory mediators.

The main results of this study are that *ATG7* mRNA and ATG7 protein hepatic expressions do not differ between patients with NW and patients presenting MO, nor among patients with NL or with NAFLD. However, patients with NASH presenting hepatic inflammation had higher *ATG7* mRNA and ATG7 protein expression in the liver than non-NASH patients. In addition, hepatic *ATG7* mRNA expression was enhanced in the NASH group compared with the SS group. Meanwhile, ATG7 protein abundance was increased in the NASH group compared with the NW group. Moreover, hepatic *ATG7* mRNA expression was increased in patients with dyslipidemia. On the other hand, we found that *ATG7* mRNA hepatic expression correlates negatively with the expression of some hepatic lipid metabolism-related genes (*CPT1a*, *LXRα,* and *RBP4*) and positively with *CB1*, *CB2,* and adiponectin hepatic expression.

For our first finding, the hepatic expression of ATG7 was not different depending on the presence of obesity. Regarding the expression of autophagy genes in adipose tissue in relation to the presence of obesity, there is controversy. On the one hand, Soussi et al. found reduced autophagy-related gene expression in adipocytes from obese patients [28]. In contrast, many authors have stated that several autophagic genes, including ATG7, are upregulated in adipocytes from subjects with obesity [29]. However, in the present study, we evaluated ATG7 expression in liver tissue according to obesity and did not find significant differences. In this case, there are no previous reports comparing this result. It has been suggested that lipid droplets or Mallory–Denk bodies accumulated in the liver could be substrates for autophagy lysosomes [30,31,32], and at the same time, autophagy tries to protect the liver against this damage [16]. Hence, ATG7-mediated autophagy seems to be influenced by liver involvement and not by obesity.

In addition, it is important to note that our patients with MO underwent a very low-calorie diet recommended before bariatric surgery [33]. Previously, it was reported in mouse models that an HFD induces an impairment of autophagy processes [17,34]. Therefore, the strict hypocaloric diet that our patients underwent could interfere with the evaluation of ATG7-mediated autophagy in the liver. In this line, it was stated that autophagy could be induced by starvation conditions [35]. Moreover, Camargo et al. demonstrated an increase in adipose tissue ATG7 expression after a diet rich in monounsaturated fats, suggesting that autophagy processes in adipose tissue may be modified by diet [36]. Perhaps a similar phenomenon can occur in liver tissue.

Later, we evaluated *ATG7* mRNA and ATG7 protein expression in the liver according to the presence of NAFLD. We did not find differential expression of ATG7 when the cohort was classified by the presence or absence of NAFLD. However, we found that *ATG7* mRNA and ATG7 protein expression levels were increased in patients presenting NASH compared to non-NASH subjects. In this regard, Singh et al. found that deletion of the *Atg7* gene in the liver of mice models increases hepatic fat content, mimicking NAFLD conditions. Thus, these authors suggested that blocking autophagy promotes hepatocellular lipid accumulation [16]. Furthermore, González-Rodríguez et al. demonstrated that hepatic autophagy is inhibited in NAFLD and NASH patients [17]. Thus, the increase in ATG7 in our NASH patients is contradictory to the few existing previous references, although it could be due to an attempt by the liver to repair liver damage through ATG7-mediated autophagy, which has been shown to eliminate excess lipids by degradation [30]. However, this is a preliminary result and would need to be validated.

Due to the previous findings, we classified our cohort in accordance with the hepatic histopathological classification to further evaluate the ATG7 implication in the pathogenesis of NAFLD. In this regard, hepatic *ATG7* mRNA and *ATG7* protein expression were enhanced in the NASH cohort compared with the SS and NW groups, respectively. This result reinforces our hypothesis, which suggests that ATG7-mediated autophagy could play a protective role in NAFLD, specifically in the NASH stage. In this sense, Hadavi et al. reported that the induction of autophagy gene expression through nutritional strategies could be useful to ameliorate a fatty liver [37].

Then, we examined the relationship between ATG7 and NASH by analyzing it according to the presence of NASH-related parameters, such as portal and lobular inflammation and hepatocellular ballooning. We observed an increase in hepatic *ATG7* mRNA and ATG7 protein expression when the cohort was classified according to the presence of global inflammation in the liver. Specifically, we found higher *ATG7* mRNA and ATG7 protein expression in women with mild lobular inflammation than in those without it. Some studies have reported that autophagy inhibits inflammation in cancerous processes [38,39]. Consistent with these studies, Baselli et al. reported that *ATG7* loss-of-function variants affected autophagy by facilitating inflammation and ballooning, two NASH characteristics [26]. These findings lead us to suppose that this potential protective role that ATG7-mediated autophagy could play in NAFLD can be also related to an attempt to prevent the inflammation that occurs in NASH [26,40]. However, more studies are needed to confirm these hypotheses.

Consequently, we studied the relationship between hepatic *ATG7* mRNA and ATG7 protein expressions, and the presence of NAFLD comorbidities, such as dyslipidemia, type 2 diabetes mellitus, hypertension, and metabolic syndrome [41]. We only found higher hepatic expression of ATG7/ATG7 in patients presenting dyslipidemia than in the cohort that did not present it. Dyslipidemia can affect hepatic fatty accumulation [42]. Thus, our finding is consistent with the fact that *ATG7* is an essential gene for autophagy, which is a homeostatic process [43]. Since subjects with dyslipidemia are more likely to have a greater accumulation of fat in the liver [42], in line with our previous results, it makes sense that autophagy is more induced in these patients, leading to the degradation of lipid droplets through lipophagy [44]. Therefore, the fact that the hepatic expression of ATG7 is increased in people with dyslipidemia could be because ATG7 is trying to regulate the impact of lipid imbalance in the liver.

Finally, we evaluated the association between ATG7 expression and other parameters related to NAFLD. Here, we found an inverse relationship between the hepatic mRNA expression of *ATG7* and the mRNA expression of some of the main lipid metabolism-related genes (*LXRα*, *CPT1a,* and *RBP4*). LXR is a crucial regulator of free fatty acid and cholesterol metabolism. It encourages de novo lipogenesis, reduces LDL catabolism, and induces hepatic steatosis [45]. On the other hand, CPT1a is a protein involved in the activation and transport of fatty acids into the mitochondria for β-oxidation [46]. Meanwhile, RBP4 is a member of the lipocalin family and the primary vitamin A that transports proteins in the blood. The liver, which hosts most of the body’s vitamin A reserves, exhibits significant levels of RBP4 expression [47]. On the one hand, Byrnes et al. described that autophagy plays a central role in lipid metabolism, regulating free fatty acid and triglyceride synthesis at the transcriptional level. In this sense, given that LXRα functions as a transcriptional regulator of cholesterol metabolism, induces de novo lipogenesis and gluconeogenesis, and is highly expressed in the liver, intestine, and fatty tissues [48,49,50], the negative association that we found with hepatic *ATG7* could make sense since this last gene is an autophagy regulator that aims to eliminate intrahepatic lipids, while *LXRα* has the opposite role by participating in hepatic lipogenesis. In addition, Kim et al. stated the inhibition of liver autophagy due to the activation of the hepatic LXRα [51]. On the other hand, Takahashi et al. reported that induction of autophagy in livers from mouse models degrades the inhibitor of LXRα, promoting its action [52]. In this regard, this association needs to be studied in depth.

Concerning CPT1a, which induces fatty acid oxidation and autophagy in the liver [53], the negative association with the hepatic expression of *ATG7*, one of the main mediators of autophagy, is logical since autophagy was found to be increased in our NASH patients, the stage when fatty acid oxidation might be inhibited [54]. However, it was found that CPT1a activation induces, at the same time, fatty acid oxidation and autophagy mechanisms. In this last-mentioned study, they evaluated mice models under HFD conditions [53], which could explain this contradiction against our human participants under a very low-calorie diet. In any case, this association also needs to be investigated.

Regarding RBP4, some studies have shown that deficiencies in vitamin A metabolism, involving RBP4, might contribute to NASH development [55,56]. In this sense, we found a negative correlation between liver *ATG7* expression and *RBP4*. Although RBP4 expression in adipose tissue has been shown to induce hepatic steatosis by promoting lipolytic pathways, RBP4 expression in the liver is not so closely related to fat accumulation itself and is more involved in the synthesis of the retinoid complex [47,57]. The negative association found in our study may be because RBP4 synthesis in hepatocytes [56] could be disrupted due to inflammation and ballooning of NASH conditions [58], specifically at the stage where ATG7-mediated autophagy is increased in our work.

Another novelty of our study is the positive relationship between *ATG7* and endocannabinoid receptor expression. The endocannabinoid system has been linked to NAFLD previously, including by our group. Specifically, we observed that CB1 increased its expression in NASH and that CB2 was related to inflammation [59]. Confirming these findings, Osei-Hyiaman et al., through an experimental study, showed that an HFD induces fatty liver mainly through the activation of hepatic CB1 receptors, and these receptors are necessary for the development of diet-induced steatosis, dyslipidemia, insulin, and leptin resistance. In addition, the CB2 receptor has been shown to play a role in modulating hepatic inflammation in a variety of studies [60,61,62,63]. Moreover, in the field of embryonic growth, activation of the endocannabinoid system has been shown to induce autophagy processes [23]. Therefore, the positive association between the hepatic expression of *ATG7* and endocannabinoid receptors is consistent with other authors since ATG7 in NASH was increased in our study.

Finally, when we studied the correlation of hepatic expression of ATG7 with pro- and anti-inflammatory molecules, we found a positive correlation with the expression of adiponectin in the liver. Adiponectin is inversely related to insulin resistance, lipid accumulation, and hepatic inflammation; it is considered an adipokine with a protective role in NAFLD [64,65]. In this sense, the positive correlation between adiponectin and hepatic expression of *ATG7* makes sense since we propose, in accordance with the obtained results, a potential protective role of ATG7-mediated autophagy in NAFLD.

This study has allowed us to evaluate the role of ATG7, the main mediator of autophagy, in NAFLD. In this case, we have found an increase in its expression in the most advanced stages of the disease, which contradicts the few existing previous studies and, therefore, we have proposed a possible protective effect of ATG7-mediated autophagy in the liver of our patients who, in addition to presenting NAFLD, have MO and followed a very restrictive calorie diet prior to bariatric surgery. However, we studied a cohort made up of only women, and NAFLD was associated with MO. In addition, whereas our MO subjects underwent a very low-calorie diet, the control group (lean subjects) followed a free diet. In this regard, our results cannot be extrapolated to men, NAFLD patients without obesity, or those having a free diet. Thus, these findings are preliminary and need to be further validated in other cohorts.

## 4. Material and Methods

### 4.1. Participants

The study was approved by the institutional review board (Institut Investigació Sanitària Pere Virgili CEIm (Comité Ético de Investigación con medicamentos, Drug Research Ethics Committee in English): 23c/2015), and all participants gave written informed consent. The study population consisted of 6 NW women (BMI < 25 kg/m^2^) and 72 women with MO (BMI ≥ 40 kg/m^2^). Liver biopsy specimens were obtained during planned laparoscopic cholecystectomy or bariatric surgery. All liver biopsies were indicated for clinical diagnosis. The exclusion criteria were as follows: (1) individuals who had alcohol consumption higher than 10 g/d; (2) patients who had acute or chronic hepatic, inflammatory, infectious, or neoplastic diseases; (3) menopausal women or women using contraceptives to avoid the interference of hormones that can cause biases in glucose and lipid metabolism, as well as in cytokine determinations; (4) women with T2DM receiving pioglitazone or insulin; and (5) patients treated with antibiotics in the previous 4 weeks.

### 4.2. Hepatopathological Diagnosis

Liver samples were scored and classified by an experienced hepatopathologist, according to Brunt’s criteria [66,67], using hematoxylin, eosin, and Masson’s trichrome stains. In this regard, women with MO were classified into NL (*n* = 11) and NAFLD (*n* = 61) groups, and these latter women were subclassified into SS (micro/macrovesicular steatosis without inflammation or fibrosis, *n* = 29) and NASH (Brunt grades 1–2, *n* = 32) subgroups. None of the NW subjects presented alterations in hepatic tissue. None of the patients with NASH in our cohort presented fibrosis.

### 4.3. Biochemical Analysis

Physical, anthropometric, and biochemical evaluations were performed on all the studied cohorts. Blood samples were extracted through a BD Vacutainer^®^ (BD IBERIA S.L., Madrid, Spain) system by specialized nurses after overnight fasting and before surgery. Venous blood samples were obtained in tubes with or without ethylenediaminetetraacetic acid, which were separated into plasma and serum aliquots by centrifugation (1507 relative centrifugal force (RCF), 4 °C, 15 min). A conventional automated analyzer was used to analyze biochemical parameters. Insulin resistance was estimated using HOMA1-IR. Cytokines, such as IL-1β, IL-6, IL-8, IL-10, IL-17, TNF-α, and omentin, were determined using multiplex sandwich immunoassays and the MILLIPLEX MAP Human Adipokine Magnetic Bead Panel 1 (HADK1MAG-61K, Millipore, Billerica, MA, USA), the MILLIPLEX MAP Human High-Sensitivity T-Cell Panel (HSTCMAG28SK, Millipore, Billerica, MA, USA), and the Bio-Plex 200 (Merck-Millipore-Life Science, Madrid, Spain) instruments, according to the manufacturer’s instructions. All these analyses were assessed at the Center for Omic Sciences (Rovira i Virgili University-Eurecat).

### 4.4. Hepatic mRNA Expression

Hepatic samples were collected during elective cholecystectomy or bariatric surgery and kept in tubes with RNAlater^®^ (#R0901, Sigma-Aldrich) (Sigma, Barcelona, Spain) at 4 °C. The samples were then processed and stored at −80 °C. An RNeasy mini kit (Qiagen Iberia S.L., Barcelona, Spain) was used to extract total RNA from the liver. Reverse transcription to cDNA was performed with the High-Capacity RNA-to-cDNA Kit (Applied Biosystems, Madrid, Spain). Real-time quantitative PCR was carried out with the TaqMan Assay predesigned by Applied Biosystems for the detection of *ATG7* (Hs00893766_m1) mRNA in the liver. We also evaluated the mRNA of some hepatic lipid metabolism-related genes, such as *SREBP1c* (Hs01088691_m1), *LXRα* (Hs00173195_m1), *FAS* (Hs00188012_m1), *CPT1a* (Hs00912671_m1), and *RBP4* (Hs00924047_m1), and endocannabinoid receptors, such as *CB1* (Hs01038522_s1), *CB2* (Hs05019229_s1), and adiponectin (Hs00977214_m1). The expression of each gene was calculated and standardized to the mRNA expression of *18S* RNA (Fn04646250_s1) after they were normalized using the control group (NW) as a reference. All reactions were duplicated in 96-well plates using the QuantStudio™ 7 Pro Real-Time PCR System (Applied Biosystem, Foster City, CA, USA).

### 4.5. Western Blot Analysis

Protein levels were evaluated in a subgroup of 21 subjects (16 MO women (5 with NL, 5 with SS, and 6 with NASH) and 5 NW subjects with NL), for whom enough tissue was available. Liver samples were homogenized in a medium containing 50 mM HEPES, 150 mM NaCl, 1 mM DTT, 0.1% SDS, and 1% protease inhibitor cocktail (Thermo Scientific, Madrid, Spain). Protein concentrations were determined by using a BCA assay kit (Thermo Scientific, Madrid, Spain). For Western blot analysis, equal amounts of protein (50 µg) were separated by SDS/PAGE (7% acrylamide) and transferred onto nylon membranes. Nonspecific binding was blocked by preincubation of the membranes with 5% (*w*/*v*) nonfat milk powder in 0.1% PBS-Tween for 1 h. Specific protein expression was detected by incubating with rabbit anti-ATG7 (Fisher Scientific SL, Madrid, Spain) antibody overnight at 4 °C, followed by incubation with an anti-rabbit IgG (GE Healthcare, Freiburg, Germany) antibody for 2 h at room temperature and developed with SuperSignal West Pico Chemiluminescent or SuperSignal Femto Maximum Sensitivity Substrate (Thermo Fisher Scientific, Waltham, MA, USA). The density of specific bands was determined by densitometry and quantified by Phoretix 1D software from TotalLab. The expression patterns of all proteins were normalized to GAPDH (Thermo Fisher Scientific, Waltham, MA, USA) and β-actin (Sigma-Aldrich, Darmstadt, Germany) liver expression.

### 4.6. Statistical Analysis

The data were analyzed using the SPSS/PC+ for Windows statistical package (version 27.0; SPSS, Chicago, IL, USA). The distribution of variables was obtained using the Kolmogorov–Smirnov test. All results are expressed as the median and the interquartile range (25th–75th). The different comparative analyses were performed using a nonparametric Mann–Whitney U test or a Kruskal–Wallis test, according to the presence of two or more groups. The coefficient of correlation (rho) between variables was calculated using Spearman’s method. *p*-values < 0.05 were considered statistically significant. Graphics were made using GraphPad Prism software (version 7.0; GraphPad, San Diego, CA, USA).

## 5. Conclusions

Although autophagy has been demonstrated in several studies to be inhibited in the development of NAFLD, in the present work, we found increased *ATG7* mRNA and ATG7 protein expression in NASH. These findings could suggest that ATG7-mediated autophagy could play a protective role in NASH, trying to counteract lipid accumulation and inflammation. However, more studies are needed to validate our hypothesis and preliminary results.

## Figures and Tables

**Figure 1 ijms-24-01324-f001:**
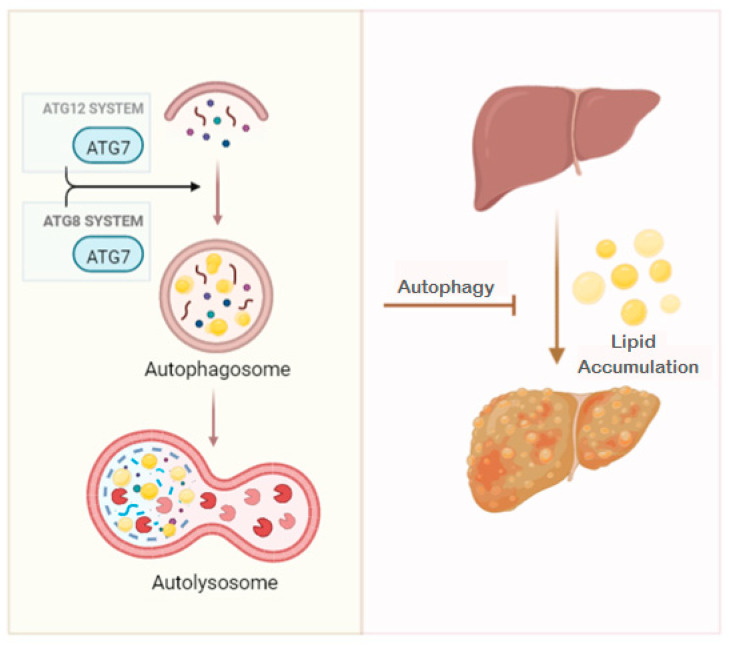
Role of ATG7 in autophagy process and in the context of metabolic-associated fatty liver (NAFLD) progression. ATG, autophagy-related gene protein.

**Figure 2 ijms-24-01324-f002:**
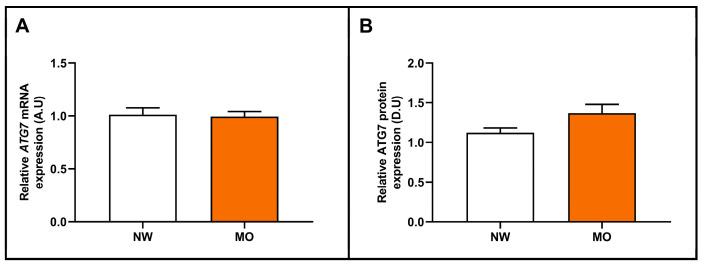
Differential relative *ATG7* mRNA (**A**) and ATG7 protein (**B**) abundance in liver samples between women with NW and MO. mRNA expression analysis: NW (*n* = 6) and MO (*n* = 72); and protein expression analysis: NW (*n* = 5) and MO (*n* = 16). NW, normal weight; MO, morbid obesity; *ATG7*/ATG7, autophagy-related 7 gene/protein; A.U, arbitrary units; D.U, densitometry units. Differences between groups were calculated using the Mann–Whitney test and *p* < 0.05 was considered statistically significant.

**Figure 3 ijms-24-01324-f003:**
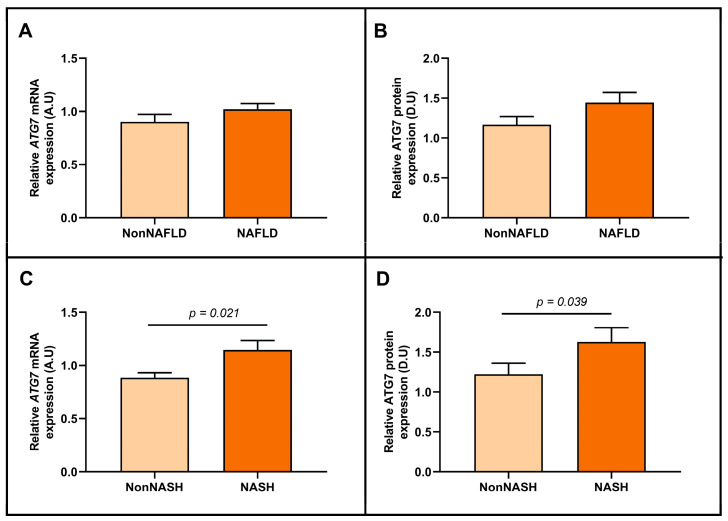
Differential relative ATG7 mRNA (**A**) and ATG7 protein (**B**) abundance in hepatic tissue between women classified according to the presence or the absence of NAFLD. Differential relative ATG7 mRNA (**C**) and ATG7 protein (**D**) abundance between women classified according to the presence or the absence of NASH. mRNA expression analysis: non-NAFLD (*n* = 17), NAFLD (*n* = 61), non-NASH (*n* = 46), and NASH (*n* = 32); and protein expression analysis: non-NAFLD (*n* = 10), NAFLD (*n* = 11), non-NASH (*n* = 15), and NASH (*n* = 6). ATG7/ATG7, autophagy-related 7 gene/protein; NAFLD, metabolic associated fatty liver disease; NASH, nonalcoholic steatohepatitis; A.U arbitrary units; D.U, densitometry units. Differences between groups were calculated using the Mann–Whitney test and *p* < 0.05 was considered statistically significant.

**Figure 4 ijms-24-01324-f004:**
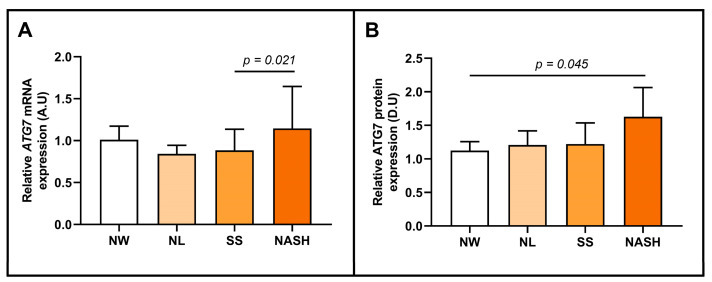
Differential relative ATG7 mRNA (**A**) and ATG7 protein (**B**) abundance in hepatic tissue between women with NW and MO subclassified as NL, SS, and NASH. mRNA expression analysis: NW (*n* = 6), NL (*n* = 11), SS (*n* = 29), and NASH (*n* = 32); and protein expression analysis: NW (*n* = 5), NL (*n* = 5), SS (*n* = 5), and NASH (*n* = 6). NW, normal weight; MO, morbid obesity; ATG7/ATG7, autophagy-related 7 gene/protein; SS, simple steatosis; NASH, nonalcoholic steatohepatitis; A.U, arbitrary units; D.U, densitometry units. Differences between groups were calculated using the Mann–Whitney test and *p* < 0.05 was considered statistically significant.

**Figure 5 ijms-24-01324-f005:**
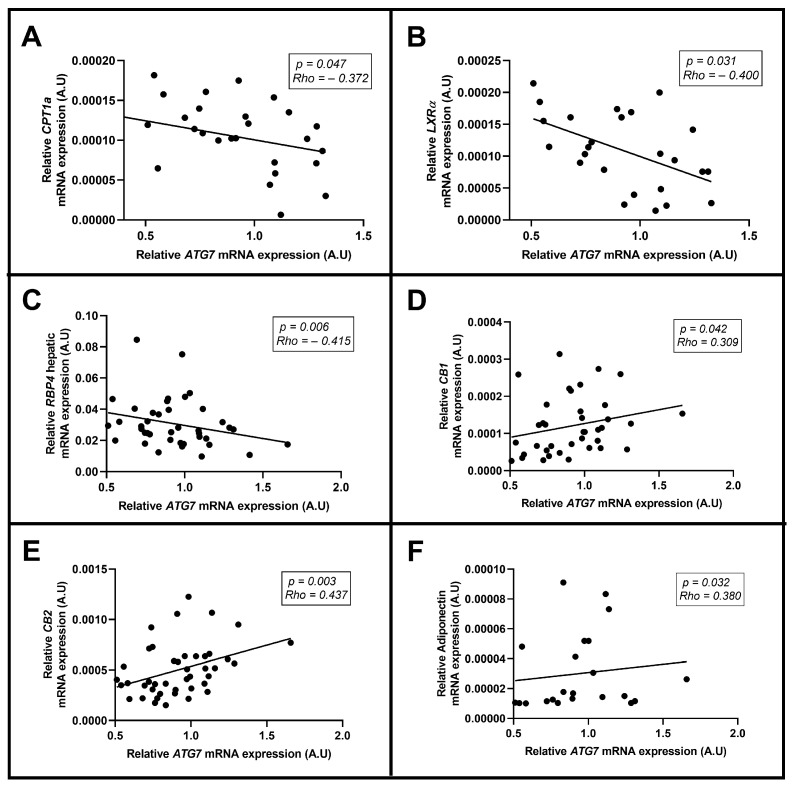
Significant correlations between *ATG7* mRNA hepatic expression and (**A**) *CPT1a*, (**B**) *LXRα*, (**C**) *RBP4*, (**D**) *CB1*, (**E**) *CB2,* and (**F**) adiponectin hepatic expression using Spearman’s (rho) correlation test. *CPT1a*, carnitine palmitoyl transferase deficiency-type 1; *LXRα*, liver X receptor alpha; *RBP4*, retinol transporter protein type 4; *CB*, cannabinoids receptors; A.U, arbitrary units. *p* < 0.05 was considered statistically significant.

**Table 1 ijms-24-01324-t001:** Anthropometric and biochemical variables of women in the studied cohort.

	MO (*n* = 72)			
Variables	NW (*n* = 6)	NL (*n* = 11)	SS (*n* = 29)	NASH (*n* = 32)
Weight (kg)	63.00 (58.00–69.00)	121.50 (107.00–130.00) *	124.00 (112.00–136.00) *	120.00 (109.90–137.20) *
BMI (kg/m^2^)	22.94 (19.93–25.00)	44.46 (49.56–42.60) *	46.36 (43.58–50.95) *	46.28 (43.26–51.04) *
SBP (mmHg)	110.00 (108.00–128.00)	125.00 (109.75–138.25)	130.00 (117.00–141.25)	130.00 (119.00–144.00) * $
DBP (mmHg)	70.00 (67.50–72.00)	68.00 (58.00–78.00)	70.00 (61.00–80.00)	73.00 (63.50–86.00) $ #
HOMA1-IR	1.50 (0.99–2.23)	2.96 (1.92–6.46)	4.28 (2.83–6.88) *	3.95 (2.58–10.70) *
Glucose (mg/dL)	84.5 (72.6–96.12)	97.11 (86.84–116.38)	108.10 (92.60–139.08)	132.06 (102.51–187.73) *$
Insulin (mUI/L)	6.41 (4.67–9.56)	13.01 (8.60–25.13) *	14.93 (10.25–27.23) *	18.91 (8.46–52.87) *
HbA1c (%)	4.45 (4.30–4.90)	5.20 (4.75–5.63)	5.60 (5.00–6.15)	5.30 (5.00–6.55)
TG (mg/dL)	123.50 (74.89–245.25)	136.00 (104.50–177.75)	188.00 (144.75–238.50) $	153.00 (119.50–197.50)
Cholesterol (mg/dL)	181.00 (134.18–197.60)	178.80 (145.45–195.75)	171.35 (154.85–189.55)	183.80 (153.50–203.00)
HDL-C (mg/dL)	40.50 (35.75–45.75)	39.50 (31.90–47.50)	37.10 (33.88–47.00)	40.00 (33.00–42.00)
LDL-C (mg/dL)	90.45 (76.48–129.40)	103.50 (86.28–123.75)	100.50 (79.15–124.55)	93.50 (83.50–128.60)
AST (UI/L)	27.50 (21.75–46.00)	36.00 (21.00–45.00)	26.00 (21.50–37.00)	40.50 (23.75–57.25)
ALT (UI/L)	19.00 (16.00–70.00)	33.00 (21.50–49.50)	29.00 (21.00–41.00)	34.00 (24.25–67.25)
GGT (UI/L)	37.00 (12.00–131.00)	19.00 (14.00–29.50)	22.50 (17.00–33.25)	26.00 (15.00–68.40)
ALP (Ul/L)	72.00 (68.50–112.50)	63.50 (50.00–73.50)	66.50 (54.00–76.00)	64.00 (55.00–77.00

MO, morbid obesity; NW, normal weight; NL, normal liver; SS, simple steatosis; NASH, nonalcoholic steatohepatitis; BMI, body mass index; SBP, systolic blood pressure; DBP, diastolic blood pressure; HOMA1-IR, homeostatic model assessment method–insulin resistance; HbA1c, glycosylated hemoglobin; TG, triglycerides; HDL-C, high-density lipoprotein cholesterol; LDL-C, low-density lipoprotein cholesterol; AST, aspartate aminotransferase; ALT, alanine aminotransferase; GGT, gamma-glutamyltransferase; ALP, alkaline phosphatase. Data are expressed as the median (interquartile range). * Significant differences between the NW group and the other groups (*p* < 0.05). $ Significant differences between the NL cohort and the other groups (*p* < 0.05). # Significant differences between the SS patients and the other women (*p* < 0.05).

## Data Availability

Data is unavailable.

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
