# Peer review of "Increased Hepatic ATG7 mRNA and ATG7 Protein Expression in Nonalcoholic Steatohepatitis Associated with Obesity"

_ijms, 2023, doi:10.3390/ijms24021324_

Round 1
Reviewer 1 Report
Very good manuscript well designed and well concluded.
Highly recommend to accept in its current format. Very important subject with clinical implications.
Author Response
Thank you very much for being part of our manuscript review committee and for your valuable comment.
Reviewer 2 Report
The manuscript entitled “The potential protective role of ATG7 in nonalcoholic steato-hepatitis associated to obesity” by Andrea Barrientos-Riosalido and colleagues describe a cohort study in which the expression of ATG7, a key autophagy regulator, both at mRNA and protein level are shown to correlate with the severity of liver disease (NASH).
The cohort is large enough and the data collected are consistent with a “possible”, although not demonstrated in the manuscript, protective role of ATG7/autophagy in disease progression.
Major issue:
- Since the data shown are interpreted in a frame of correlation rather than causality, I would suggest the authors to slightly downgrade some statements about the protective role of ATG7.
- No Western blot data are shown but only quantification. Please add representative WB with ATG7 and other autophagy markers (SQSTM1, LC3) to evaluate the autophagy pathway in terms of flux and lysosomal degradation of autophagosomes.
- Please expand the discussion on the possible transcriptional link between ATG7 (that is not a transcription factor), the autophagy pathway and the expression of genes such as CPT1, LXR, CB1 and CB2
- The authors should add data (either mRNA or WB) about the correlation of ATG7 level and the level of Lipid droplets markers such as PLIN2 is the different classes of individuals included in this study.
Reviewer 3 Report
The manuscript is written well.
Author Response

(The authors gave the same response as above.)

Round 2
Reviewer 2 Report
The authors have replied to all my comments. I understand that the limited budget and scarcity of biological samples are the main causes of the lack of any further experiments. I acknowledge the authors have rewritten some parts of the manuscript, downgrading some statements.
Author Response
Thank you for your positive feedback on our review.